# Three Iterations of $(d − 1)$-WL Test Distinguish Non Isometric Clouds of $d$-dimensional Points

**Valentino Delle Rose**[1,2]**, Alexander Kozachinskiy**[1,3]**, Cristóbal Rojas**[1,2]

**Mircea Petrache**[1,2,4]**, Pablo Barceló**[1,2,3]

[1] Centro Nacional de Inteligencia Artificial, Chile
[2] Instituto de Ingeniería Matemática y Computacional, Universidad Católica de Chile
[3] Instituto Milenio Fundamentos de los Datos, Chile
[4] Departamento de Matemática, Universidad Católica de Chile
valentino.dellerose@cenia.cl, alexander.kozachinskyi@cenia.cl
cristobal.rojas@mat.uc.cl, mpetrache@mat.uc.cl, pbarcelo@uc.cl

## Abstract

The Weisfeiler–Lehman (WL) test is a fundamental iterative algorithm for checking the isomorphism of graphs. It has also been observed that it underlies the design of several graph neural network architectures, whose capabilities and performance can be understood in terms of the expressive power of this test. Motivated by recent developments in machine learning applications to datasets involving three-dimensional objects, we study when the WL test is *complete* for clouds of Euclidean points represented by complete distance graphs, i.e., when it can distinguish, up to isometry, any arbitrary such cloud. Our main result states that the $(d − 1)$-dimensional WL test is complete for point clouds in $d$-dimensional Euclidean space, for any $d \geq 2$, and only three iterations of the test suffice. Our result is tight for $d = 2, 3$. We also observe that the $d$-dimensional WL test only requires one iteration to achieve completeness.

## 1 Introduction

**Context.** Recent work in machine learning has raised the need to develop effective and efficient tests for checking if two three-dimensional point clouds, i.e., finite sets of points in $\mathbb{R}^3$, are *isometric* [15, 9, 2, 12]. Recall that, given two such point clouds $P$ and $Q$, an isometry is a distance-preserving bijection between the points in $P$ and $Q$. The importance of these tests is that they provide the foundations for designing neural network architectures on point clouds that are capable of fully exploiting the structure of the data [18, 14]. It has been observed that the *incompleteness* of any such an architecture, i.e., the inability to recognize a point cloud up to isometry, can affect its learning performance [16]. Understanding which is the simplest test that allows detecting isometries in this scenario is thus essential not only for developing "complete" architectures but also to make them as efficient as possible in terms of the computational resources they need to use.

Point clouds can be represented as complete graphs in which each edge is labeled with the distance between the corresponding points. Under this representation, detecting the isometry of two point clouds is reduced to detecting an isomorphism between their graph representation. Not surprisingly, then, much of the work on developing so-called *geometric* tests for detecting isometries over point clouds is inspired by the literature on isomorphism tests from graph theory. Of particular importance

37th Conference on Neural Information Processing Systems (NeurIPS 2023).

in this context has been the use of geometric versions of the well-known *Weisfeiler-Lehman test* (WL test) for graph isomorphism [17].

Intuitively, the $\ell$-dimensional geometric WL test ($\ell$-WL test), for $\ell \geq 1$, iteratively colors each tuple $\bar{v}$ of $\ell$ points in a point cloud. The color of $\bar{v}$ in round 0 is a complete description of the mutual distances between the points that belong to the tuple. In round $t + 1$, for $t \geq 0$, the color of $\bar{v}$ is updated by combining in a suitable manner its color in iteration $t$ with the color of each one of its *neighbors*, i.e., the tuples $\bar{v}'$ that are obtained from $\bar{v}$ by exchanging exactly one component of $\bar{v}$ with another point in the cloud. The dimensionality of the WL test is therefore a measure of its computational cost: the higher the $\ell$, the more costly it is to implement the $\ell$-dimensional WL test.

For checking if two point clouds are isometric, the geometric WL test compares the resulting color patterns. If they differ, then we can be sure the point clouds are not isometric (that is, the test is *sound*). An important question, therefore, is what is the minimal $\ell \geq 1$ for which the geometric $\ell$-WL test is *complete*, i.e., the fact that the color patterns obtained in two point clouds are the same implies that they are isometric.

There has been important progress on this problem recently: (a) Pozdnyakov and Ceriotti have shown that the geometric 1-WL test is incomplete for point clouds in 3D; that is, there exist isometric point clouds in three dimensions that cannot be distinguished by the geometric 1-WL test [15]; (b) Hordan et al. have proved that 3-WL test is complete in 3D after 1 iteration when initialized with Gram matrices of the triples of points instead of the mutual distances in these triples. A similar result has recently been obtained in [12]. Hordan et al. also gave a complete "2-WL-like" test, but this test explicitly uses coordinates of the points.

As further related results, in [10], geometric WL tests have been compared to the expressivity of invariant and equivariant graph neural networks. Non-geometric related results include e.g. [5], where for explicit graphs on $n$ vertices it is shown that $\ell$-WL requires $O(n)$ iterations for distinguishing them, whereas $2\ell$-WL requires $O(\sqrt{n})$ and $(3\ell - 1)$-WL requires $O(\log n)$ iterations, and [19], which has a promising proposal for generalized distances on non-geometric graphs, based on biconnectivity.

**Main theoretical results.** Our previous observation show an evident gap in our understanding of the problem: What is the minimum $\ell$, where $\ell = 2, 3$, for which the geometric $\ell$-dimensional WL test is complete over three-dimensional point clouds? Our main contributions are the following:

- We show that for any $d > 1$ the geometric $(d - 1)$-WL test is complete for detecting isometries over point clouds in $\mathbb{R}^d$. This is the positive counterpart of the result in [14] (namely, that geometric 1-WL test is incomplete in dimension $d = 3$), by showing that geometric 1-WL is complete for $d = 2$ and that geometric 2-WL test is complete for $d = 3$. Further, only three rounds of the geometric $(d - 1)$-WL test suffice to obtain this completeness result.

- We provide a simple proof that a single round of the geometric $d$-WL test is sufficient for identifying point clouds in $\mathbb{R}^d$ up to isometry, for each $d \geq 1$. This can be seen as a refinement of the result in [9], with the difference that our test is initialized with the mutual distances inside $d$-tuples of points (as in the classical setting) while theirs is initialized with Gram matrices of $d$-tuples of points. In other words, the initial coloring of [9], for each $d$-tuple of points, in addition to their pairwise distances, includes their distances to the *origin*, while in our result we do not require this additional information.

These results, as well as previously mentioned results obtained in the literature, are all based on the standard *folklore* version of the $\ell$-WL test (as defined, e.g., in [3]). This is important because another version of the test, known as the *oblivious* $\ell$-WL test, has also been studied in the machine learning literature [14, 13]. It is known that, for each $\ell \geq 1$, the folklore $\ell$-WL test has the same discriminating power as the oblivious $(\ell + 1)$-WL test [7].

Table 1 summarizes the state of the art concerning the distinguishing power of the geometric $\ell$-WL test; note that it is not known whether the $(d - 2)$-WL test is incomplete for $d \geq 4$.

**Relationship with graph neural networks.** Graph Neural Networks (GNNs) are specialized neural networks designed to process data structured as graphs [8, 11]. Among GNNs, *Message Passing* GNNs (MPGNNs) use a message-passing algorithm to disseminate information between nodes in a graph [6]. The relationship between the 1-WL test and MPGNNs is now a fundamental subject in

| Is $\ell$-WL complete for $\mathbb{R}^d$? | | |
| --- | --- | --- |
| | 1-WL | 2-WL | 3-WL |
| $\mathbb{R}^2$ | **Complete** in 3 iterations Theorem 2.1 | **Complete** in 1 iteration Theorem 5.1 | |
| $\mathbb{R}^3$ | **Incomplete** [14] | **Complete** in 3 iterations Theorem 2.1 | **Complete** in 1 iteration Theorem 5.1 |
| $\mathbb{R}^4$ | | **Open** | **Complete** in 3 iterations Theorem 2.1 |
| | $(d-2)$-WL | $(d-1)$-WL | $d$-WL |
| $\mathbb{R}^d$ | **Open** | **Complete** in 3 iterations Theorem 2.1 | **Complete** in 1 iteration Theorem 5.1 |

Table 1: What is known about the distinguish power of geometric $\ell$-WL.

this field. Seminal research has shown that these two approaches are essentially equivalent in their ability to distinguish non-isomorphic graph pairs [14, 18]. Additionally, Morris et al. [14] proposed *higher-dimensional* $\ell$-MPGNNs that have the same discriminating power as the oblivious $\ell$-WL test, and hence as the folklore $(\ell - 1)$-WL test, for $\ell > 1$.

The geometric WL test studied here corresponds to a particular case of the *relational* WL test, i.e., a suitable version of the WL test that is tailored for edge-labeled graphs. Connections between the relational WL test and so-called *relational* MPGNNs have recently been established by [1]. In particular, relational $\ell$-MPGNNs have the same discriminating power as the (folklore) geometric $(\ell-1)$-WL test, for $\ell > 1$. Our main result implies then that there is no need to create specialized GNN architectures for distinguishing non-isometric point clouds in $\mathbb{R}^d$. Instead, relational $d$-MPGNNs are sufficient for this task.

Experimental evaluation of MPGNNs, based on the folklore 2-WL test, on data sets from molecular physics, and their comparison with state-of-art models, was recently performed in [12].

## 2 Formal Statement of the Main Result

Consider a cloud of $n$ points $S = \{p_1, \ldots, p_n\}$ in $\mathbb{R}^d$. We are interested in the problem of finding representations of such clouds that completely characterize them up to isometries, while at the same time being efficient from an algorithmic point of view. Our main motivation is to understand the expressiveness of the WL algorithm when applied to point clouds in euclidean space seen as complete distance graphs. Let us briefly recall how this algorithm works.

A function whose domain is $S^\ell$ will be called an $\ell$-*coloring* of $S$. The $\ell$-WL algorithm is an iterative procedure which acts on $S$ by assigning, at iteration $i$, an $\ell$-coloring $\chi_{\ell,S}^{(i)}$ of $S$.

**Initial coloring.** The initial coloring, $\chi_{\ell,S}^{(0)}$, assigns to each $\ell$-tuple $\mathbf{x} = (x_1, \ldots, x_\ell) \in S^\ell$ the color $\chi_{\ell,S}^{(0)}(\mathbf{x})$ given by the $\ell \times \ell$ matrix

$$\chi_{\ell,S}^{(0)}(\mathbf{x})_{ij} = d(x_i, x_j) \quad i, j = 1, \ldots, \ell$$

of the relative distances inside the $\ell$-tuple (for $\ell = 1$ we have a trivial coloring by the $0$ matrix).

**Iterative coloring.** At each iteration, the $\ell$-*WL algorithm* updates the current coloring $\chi_{\ell,S}^{(i)}$ to obtain a refined coloring $\chi_{\ell,S}^{(i+1)}$. The update operation is defined slightly differently depending on whether $\ell = 1$ or $\ell \geq 2$.

- For $\ell = 1$, we have:

$$\chi_{1,S}^{(i+1)}(x) = \left( \chi_{1,S}^{(i)}(x), \{\!\{ (d(x,y), \chi_{1,S}^{(i)}(y)) \mid y \in S \}\!\} \right).$$

  That is, first, $\chi_{i+1}(x)$ remembers the coloring of $x$ from the previous step. Then it goes through all points $y \in S$. For each $y$, it stores the distance from $x$ to $y$ and also the coloring

of $y$ from the previous step, and it remembers the multiset of these pairs. Note that one can determine which of these pairs comes from $y = x$ itself since this is the only point with $d(x, y) = 0$. We also note that $\chi_{1,S}^{(1)}(x)$ corresponds to the multiset of distances from $x$ to the points of $S$.

- To define the update operation for $\ell \geq 2$, we first introduce additional notation. Let $\mathbf{x} = (x_1, \ldots, x_\ell) \in S^\ell$ and $y \in S$. By $\mathbf{x}[y/i]$ we mean the tuple obtained from $\mathbf{x}$ by replacing its $i$th coordinate by $y$. Then the update operation can be defined as follows:

$$\chi_{\ell,S}^{(i+1)}(\mathbf{x}) = \left( \chi_{\ell,S}^{(i)}(\mathbf{x}), \{\!\!\{ \left( \chi_{\ell,S}^{(i)}(\mathbf{x}[y/1]), \ldots, \chi_{\ell,S}^{(i)}(\mathbf{x}[y/\ell]) \right) \mid y \in S \}\!\!\} \right). \tag{1}$$

In other words, first, $\chi_{\ell,S}^{(i+1)}(\mathbf{x})$ remembers the coloring of $\mathbf{x}$ from the previous step, as before. Then, it goes through all $y \in S$ and considers the $\ell$ tuples $\mathbf{x}[y/1], \ldots, \mathbf{x}[y/\ell]$. It then takes the colorings of these tuples from the previous step and puts them into a tuple. The new coloring now remembers the multiset of these tuples.

In this paper we show that the coloring obtained after 3 iterations of $(d-1)$-WL is a complete isometry invariant for point clouds in $\mathbb{R}^d$. More precisely, we show:

**Theorem 2.1** (Main Theorem). *For any $d \geq 2$ and for any finite set $S \subseteq \mathbb{R}^d$, the following holds. Let $\chi_{d-1,S}^{(3)}$ be the coloring of $S^{d-1}$ obtained after 3 iterations of the $(d-1)$-WL algorithm on the distance graph of $S$. Then, knowing the multiset*

$$\mathcal{M}_{d-1}^{(3)}(S) = \{\!\!\{ \chi_{d-1,S}^{(3)}(\mathbf{s}) \mid \mathbf{s} \in S^{d-1} \}\!\!\},$$

*one can determine $S$ up to an isometry.*

Our proof is constructive in the sense that we exhibit an algorithm which, upon input $\mathcal{M}_{d-1}^{(3)}(S)$, computes the coordinates of a point cloud $S'$ which is isometric to $S$. In particular, if $\widetilde{S} \subset \mathbb{R}^d$ is another point cloud such that $\mathcal{M}_{d-1}^{(3)}(\widetilde{S}) = \mathcal{M}_{d-1}^{(3)}(S)$, then $S$ and $\widetilde{S}$ are isometric.

Our result is also true for point clouds that are *multisets* $S \subseteq \mathbb{R}^d$ (for which distance graphs can have edges, labeled by $0$, connecting two nodes, representing the same point in space), but for simplicity, we now present the argument for sets (although no significant changes are needed).

**Organization of the paper.** We start in Section 3 with the proof of the two-dimensional case, which while somewhat simpler, will allow us to introduce the general strategy. In Section 4 we explain how to implement this strategy in general for $d > 2$. Then, in Section 5, we discuss the completeness of one round of $d$-WL. Finally, in Section 6 we present open problems and limitations. Due to space constraints, some proofs have been relegated to the appendix.

## 3 Three iterations of 1-WL distinguish clouds in the plane

Let $S \subseteq \mathbb{R}^2$ be a cloud of $n$ points in the plane. Our task is to reconstruct $S$ up to an isometry, using as input the information contained in $\chi_{1,S}^{(3)}$. This means to find a point cloud $S'$ in the plane which is an image of $S$ under some isometry. Our proof has two main steps: *Initialization* and *Reconstruction*. In the Initialization Step we show how to extract from $\chi_{1,S}^{(3)}$ the relevant information we need, which we call *initialization data*. In the Reconstruction step, we describe an algorithm that, given some initialization data, computes the coordinates of the desired isometric cloud.

In our reconstruction algorithm, we employ the notion of the *barycenter* of a point cloud (also known as the center of mass), which we denote by $b$, and is defined by:

$$b = \frac{1}{n} \sum_{w \in S} w.$$

For simplicity, we translate $S$ by $-b$ so that the new barycenter sits at $b = 0$. Notice that since our reconstruction is up to isometries, this assumption does not affect the gerenality of our result. For each $w \in S$, let $\|w\|$ denote its norm (its distance to $b = 0$).

We say that two points $u, v \in S$ satisfy the *cone condition* if $u \neq 0$, $v \neq 0$, and, moreover,

- if $0, u, v$ lie on the same line, then all points of $S$ lie on this line;
- if $0, u, v$ do not lie on the same line, then the interior of
  $\mathsf{Cone}(u, v) = \{\alpha u + \beta v : \alpha, \beta \in [0, +\infty)\}$ does not contain points from $S$ (see the first picture on Figure 1, the red area between $(0, u)$ and $(0, v)$ is disjoint from $S$).

In order to initialize our reconstruction algorithm, we need the following information about $S$. We assume that $S$ has more than 1 point (otherwise there is nothing to do).

***Initialization Data***: the initialization data consists of a real number $d_0 \geq 0$ and two multisets $M, M'$ such that for some $u, v \in S$, satisfying the cone condition, it holds that $d_0 = d(u, v)$ and

$$M = M_u = \{\!\{(d(u, y), \|y\|) : y \in S\}\!\}; \quad M' = M_v = \{\!\{(d(v, y), \|y\|) : y \in S\}\!\}.$$

We will start by describing the Reconstruction Algorithm, assuming that the initialization data is given. We will then show how to extract this data from $\chi_{1,S}^{(3)}$ in the Initialization Step bellow.

***Reconstruction algorithm.*** Assume that initialization data $(d_0, M, M')$ is given. Our task is to determine $S$ up to isometry. Note that from $M$ we can determine $\|u\|$. Indeed, in $M$ there exists exactly one element whose first coordinate is 0, and this element is $(0, \|u\|)$. Likewise, from $M'$ we can determine $\|v\|$. We are also given $d_0 = d(u, v)$. Overall, we have all the distances between $0$, $u$, and $v$. Up to a rotation of $S$, there is only one way to put $u$, it has to be somewhere on the circle of radius $\|u\|$, centered at the origin. We fix any point of this circle as $u$. After that, there are at most two points where we can have $v$. More specifically, $v$ belongs to the intersection of two circles: one of radius $\|v\|$ centered at the origin, and the other of radius $d(u, v)$ centered at $u$. These two circles are different (remember that the cone condition includes a requirement that $u \neq 0$). Hence, they intersect by at most two points. These points are symmetric w.r.t. the line that connects the centers of the two circles, i.e., $0$ and $u$. Thus, up to a reflection through this line (which preserves the origin and $u$), we know where to put $v$.

Henceforth, we can assume the coordinates of $u$ and $v$ are known to us. Note that so far, we have applied to $S$ a translation (to put the barycenter at the origin), a rotation (to fix $u$), and a reflection (to fix $v$). We claim that, up to this isometry, $S$ can be determined uniquely.

Let $\mathsf{Refl}_u$ and $\mathsf{Refl}_v$ denote the reflections through the lines $(0, u)$ and $(0, v)$, respectively. We first observe that from $M$ we can restore each point of $S$ up to a reflection through the line $(0, u)$. Likewise, from $M'$ we can do the same with respect to the line $(0, v)$. More precisely, we can compute the following multisets:

$$L_u = \{\!\{\{y, \mathsf{Refl}_u y\} \mid y \in S\}\!\}, \qquad L_v = \{\!\{\{y, \mathsf{Refl}_v y\} \mid y \in S\}\!\}.$$

Indeed, consider any $(d(u, y), \|y\|) \in M$. What can we learn about $y \in S$ from this pair of numbers? These numbers are distances from $y$ to $u$ and to $0$. Thus, $y$ must belong to the intersection of two circles: one with the center at $u$ and radius $d(u, y)$ and the other with the center at $0$ and radius $\|y\|$. Again, since $u \neq 0$, these two circles are different. Thus, we obtain at most two points $z_1, z_2$ where one can put $y$. We will refer to these points as *candidate locations* for $y$ w.r.t. $u$. They can be obtained one from the other by the reflection $\mathsf{Refl}_u$ through the line connecting $0$ and $u$. Hence, $\{z_1, z_2\} = \{y, \mathsf{Refl}_u y\}$. To compute $L_u$, we go through $(d(u, y), \|y\|) \in M$, compute candidate locations $z_1, z_2$ for $y$, and put $\{z_1, z_2\}$ into $L_u$. In a similar fashion, one can compute $L_v$ from $M'$.

Let us remark that elements of $L_u$ and $L_v$ are sets of size 2 or 1. A set of size 2 appears as an element of $L_u$ when some $y$ has two distinct candidate locations w.r.t. $u$, that is when $y$ does not lie on the line $(0, u)$. In turn, when $y$ does lie on this line, we have $z_1 = z_2 = y$ for two of its candidate locations, giving us an element $\{y\} \in L_u$, determining $y$ uniquely. The same thing happens with respect to $L_v$ for points that lie on the line $(0, v)$,

The idea of our reconstruction algorithm is to gradually exclude some candidate locations so that more and more points get a unique possible location. What allows us to start is that $u$ and $v$ satisfy the cone condition; this condition gives us some area that is free of points from $S$ (thus, one can exclude candidates belonging to this area).

The easy case is when $0, u$, and $v$ belong to the same line. Then, by the cone condition, all points of $S$ belong to this line. In this case, every point of $S$ has just one candidate location. Hence, both multisets $L_u$ and $L_v$ uniquely determine $S$.

Assume now that $0, u$, and $v$ do not belong to the same line. As in the previous case, we can uniquely restore all points that belong to the line connecting $0$ and $u$, or to the line connecting $0$ and $v$ (although now these are two different lines). Indeed, these are points that have exactly one candidate location w.r.t. $u$ or w.r.t. $v$. They can be identified by going through $L_u$ and $L_v$ (we are interested in points $z$ with $\{z\} \in L_u \cup L_v$).

The pseudocode for our reconstruction algorithm is given in Algorithm 1. We now give its verbal description. Let us make a general remark about our algorithm. Once we find a unique location for some $y \in S$, we remove it from our set in order to reduce everything to the smaller set $S \setminus \{y\}$. This is implemented by updating the multisets $L_u$ and $L_v$ so that $y$ is not taken into account in them. For that, we just remove $\{y, \mathsf{Refl}_u y\}$ from $L_u$ and $\{y, \mathsf{Refl}_v y\}$ from $L_v$ (more precisely, decrease their multiplicities by 1).

From now on, we assume that these two lines (connecting $0$ and $u$, and $0$ and $v$, respectively) are free of the points of $S$. These lines contain the border of the cone $\mathsf{Cone}(u, v)$. At the same time, the interior of this cone is disjoint from $S$ due to the cone condition. Thus, in fact, the whole $\mathsf{Cone}(u, v)$ is disjoint from $S$.

We now go through $L_u$ and $L_v$ in search of points for which one of the candidate locations (either w.r.t. $u$ or w.r.t. $v$) falls into the "forbidden area", that is, into $\mathsf{Cone}(u, v)$. After restoring these points and deleting them, we notice that the "forbidden area" becomes larger. Indeed, now in $S$ there are no points that fall into $\mathsf{Cone}(u, v)$ under one of the reflections $\mathsf{Refl}_u$ or $\mathsf{Refl}_v$. In other words, the updated "forbidden area" is $F = \mathsf{Cone}(u, v) \cup \mathsf{Refl}_u \mathsf{Cone}(u, v) \cup \mathsf{Refl}_v \mathsf{Cone}(u, v)$. If the absolute angle between $u$ and $v$ is $\alpha_{uv}$, then, $F$ has total amplitude $3\alpha_{uv}$. We now iterate this process, updating $F$ successively. At each step, we know that after all removals made so far, $S$ does not have points in $F$. Thus, points of $S$ that fall into $F$ under $\mathsf{Refl}_u$ or under $\mathsf{Refl}_v$ can be restored uniquely. After deleting them, we repeat the same operation with $F \cup \mathsf{Refl}_u F \cup \mathsf{Refl}_v F$ in place of $F$.

---

**Algorithm 1:** Reconstruction algorithm

---

1   $S := \varnothing$;

2   **for** $\{z\} \in L_u \cup L_v$ **do**

      // Restoring points from the lines $(0, u)$ and $(0, v)$

3      Put $z$ into $S$;

4      Remove $\{z\}$ from $L_u \cup L_v$;

5   **end for**

6   $F := \mathsf{Cone}(u, v)$;

7   **while** $F \neq \mathbb{R}^2$ **do**

8      **for** $\{z_1, z_2\} \in L_u \cup L_v$ **do**

         // Restoring points that have one candidate location in the forbidden area

9        **if** $z_1 \in F$ *or* $z_2 \in F$ **then**

10          Set $y = z_1$ if $z_2 \in F$ and $y = z_2$ if $z_1 \in F$;

11          Put $y$ into $S$;

12          Remove $\{y, \mathsf{Refl}_u y\}$ from $L_u$ and $\{y, \mathsf{Refl}_v y\}$ from $L_v$;

13      **end for**

      // Updating forbidden area

14      $F = F \cup \mathsf{Refl}_u F \cup \mathsf{Refl}_v F$;

15   **end while**

16   Output $S$;

---

After $k$ such "updates", $F$ will consist of $2k + 1$ adjacent angles, each of size $\alpha_{uv}$, with $\mathsf{Cone}(u, v)$ being in the center. In each update, we replace $F$ with $F \cup \mathsf{Refl}_u F \cup \mathsf{Refl}_v F$. This results in adding two angles of size $\alpha_{uv}$ to both sides of $F$. Indeed, if we look at the ray $(0, u)$, it splits our current $F$ into two angles, one of size $(k + 1)\alpha_{uv}$ and the other of size $k\alpha_{uv}$. Under $\mathsf{Refl}_u$, the part whose size

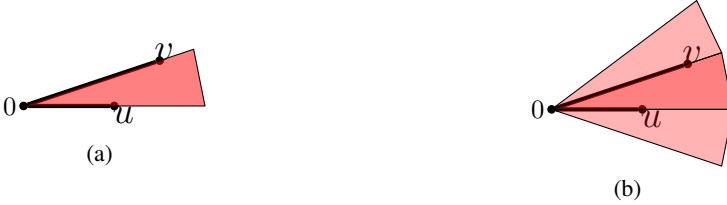

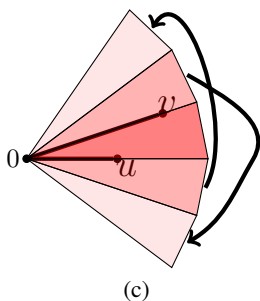

(c)

Figure 1: Growth of the "forbidden" area.

is $(k+1)\alpha_{uv}$ adds an angle of size $\alpha_{uv}$ to the other part. Analogously, $\text{Refl}_v$ adds an angle of size $\alpha_{uv}$ from the opposite side of $F$. See Figure 1 for the illustration of this process.

Within at most $1 + \frac{\pi}{\alpha_{uv}}$ such steps, $F$ covers all angular directions, thus completing the reconstruction of $S$.

***Initialization step.*** We explain how to obtain the Initialization Data about $S$ from $\mathcal{M}_1^{(3)}(S)$.

We start by observing that from the first iteration of 1-WL, we can compute $\|x\|$ for all $x \in S$. As the following lemma shows, this holds in any dimension, with the same proof. We temporarily omit the current hypothesis $b = 0$, in order to use the lemma later without this hypothesis.

**Lemma 3.1** (The Barycenter Lemma). *Take any $n$-point cloud $S \subseteq \mathbb{R}^d$ and let*

$$D_x = \{\!\{ d(x,y) \mid y \in S \}\!\}.$$

*Then for every $x \in S$, knowing $D_x$ and the multiset $\{\!\{ D_y \mid y \in S \}\!\}$, one can determine the distance from $x$ to the barycenter of $S$.*

*Proof.* Consider the function $f \colon \mathbb{R}^d \to [0, +\infty)$ defined as $f(x) = \sum_{y \in S} \|x - y\|^2$, namely $f(x)$ is the sum of the squares of all elements of $D_x$ (with multiplicities). It follows that $\sum_{y \in S} f(y)$ is determined by $\{\!\{ D_y \mid y \in S \}\!\}$. The lemma is thus proved if we prove the following equality

$$\|x - b\|^2 = \frac{1}{n} \left( f(x) - \frac{1}{2n} \sum_{y \in S} f(y) \right). \tag{2}$$

To prove the above, we first write:

$$
\begin{aligned}
f(x) = \sum_{y \in S} \|x - y\|^2 &= \sum_{y \in S} \|(x - b) + (b - y)\|^2 \\
&= \sum_{y \in S} \left( \|x - b\|^2 + 2\langle x - b, b - y \rangle + \|b - y\|^2 \right) \\
&= n \cdot \|x - b\|^2 + 2\langle x - b, n \cdot b - \sum_{y \in S} y \rangle + \sum_{y \in S} \|b - y\|^2 \\
&= n \cdot \|x - b\|^2 + \sum_{y \in S} \|b - y\|^2 \quad \text{(by definition of barycenter)}.
\end{aligned}
$$

Denote $\Gamma = \sum_{y \in S} \|b - y\|^2$. Substituting the expression for $f(x)$ and $f(y)$ from above into the right-hand side of (2), we get:

$$\frac{1}{n}\left(f(x) - \frac{1}{2n}\sum_{y \in S} f(y)\right) = \frac{1}{n}\left(n \cdot \|x - b\|^2 + \Gamma - \frac{1}{2n}\sum_{y \in S}\left(n \cdot \|y - b\|^2 + \Gamma\right)\right)$$

$$= \frac{1}{n}\left(n \cdot \|x - b\|^2 + \Gamma - \frac{1}{2n}(n\Gamma + n\Gamma)\right) = \|x - b\|^2,$$

as required. $\qquad\qquad\square$

By definition, $\chi_{1,S}^{(1)}(x)$ determines the multiset $D_x = \{\!\{d(x, y) \mid y \in S\}\!\}$ of distances from $x$ to points of $S$. Since we are given the multiset $\mathcal{M}_1^{(3)}(S)$, we also know the multset $\mathcal{M}_1^{(1)}(S) = \{\!\{\chi_{1,S}^{(1)}(x) \mid x \in S\}\!\}$ (labels after the third iterations determine labels from previous iterations). In particular, this gives us the multiset $\{\!\{D_x \mid x \in S\}\!\}$. Overall, due to the Barycenter lemma, we conclude that $\chi_{1,S}^{(1)}(x)$ can be converted into $\|x\|$.

Now, remember that

$$\chi_{1,S}^{(2)}(x) = \left(\chi_{1,S}^{(1)}(x), \{\!\{(d(x, y), \chi_{1,S}^{(1)}(y)) \mid y \in S\}\!\}\right).$$

By converting $\chi_{1,S}^{(1)}(y)$ into $\|y\|$ for all $y \in S$ here, one can convert $\chi_{1,S}^{(2)}(x)$ into the multiset

$$M_x = \{\!\{(d(x, y), \|y\|) \mid y \in S\}\!\}.$$

We need one more iteration to find $d(u, v), M_u, M_v$ for some $u, v \in S$ satisfying the cone condition. In fact, we only need

$$\chi_{1,S}^{(3)}(u) = \left(\chi_{1,S}^{(2)}(u), \{\!\{\left(d(u, y), \chi_{1,S}^{(2)}(y)\right): y \in S\}\!\}\right)$$

for arbitrary $u \in S$ with $u \neq 0$. Since $\chi_{1,S}^{(3)}(u)$ determines $\|u\|$, such $\chi_{1,S}^{(3)}(u)$ can indeed be selected from $\mathcal{M}_1^{(3)}(S)$ (and since we assume that $S$ has more than one point, we know that there are points in $S$ that are different from 0).

Due to the fact that $\chi_{1,S}^{(2)}(y)$ can be converted to $M_y$, we can in turn convert $\chi_{1,S}^{(3)}(u)$ into the multiset $\mathcal{A} = \{\!\{\left(d(u, y), M_y\right): y \in S\}\!\}$. In particular, since $y = u$ is the only point for which $d(u, y) = 0$, we can compute $M_u$ from $\mathcal{A}$. Once we have $M_u$, the rest of the initialization goes as follows. First note that for a given element $(d(u, y), M_y)$ in $\mathcal{A}$ with $d(u, y) > 0$ (so that $y \neq u$), we can look in $M_y$ for the only element with 0 as the first entry, whose second entry is then $\|y\|$. So we can obtain $(d(u, y), \|y\|)$. As in the Reconstruction Algorithm, we then have only two possibilities for the location of $y$ relative to $u$, say $y_1$ and $y_2 = \mathsf{Refl}_u(y_1)$. It follows that the absolute value of the angle $\alpha_{uy}$ between $u$ and $y$ is uniquely determined (if $\|y\| = 0$, we set $\alpha_{uy} = 0$), and we can compute it from $\mathcal{A}$. In order to select $v$, we go though $\mathcal{A}$ and look for $v$ such that $\alpha_{uv}$ is the smallest angle among $\{\alpha_{uy} \mid y \in S, 0 < \alpha_{uy} < \pi\}$. If such a $v$ does not exist, all points of $S$ must lie on the line connecting 0 and $u$. In this case, the cone condition is satisfied, for example, for $u$ and $v = u$. Thus, we can initialize with $d_0 = 0, M = M' = M_u$. If $v$ as above exists, there can be no point in the interior of $\mathsf{Cone}(u, v)$, since otherwise there would be $y$ with $0 < \alpha_{uy} < \alpha_{uv} < \pi$, contradicting the minimality of $\alpha_{uv}$. Thus, the cone condition is satisfied for $u, v$. We can then set $d_0 = d(u, v)$, $M = M_u$ and $M' = M_v$.

## 4 Proof of Main Theorem for $d > 2$

We now present the proof for the case $d > 2$. The strategy of the proof has the same structure as for $d = 2$. Since the objects involved now are more general, it will be convenient to introduce some terminology. Let $\mathbf{x} = (x_1, \ldots, x_k) \in (\mathbb{R}^d)^k$ be a $k$-tuple of points in $\mathbb{R}^d$. The ***distance matrix*** of $\mathbf{x}$ is the $k \times k$ matrix $A$ given by $A_{ij} = d(x_i, x_j), i, j = 1, \ldots, k$.

Now, let $S \subseteq \mathbb{R}^d$ be a finite set. Then the ***distance profile*** of $\mathbf{x}$ w.r.t. $S$ is the multiset

$$D_{\mathbf{x}} = \{\!\{ \big( d(x_1, y), \ldots, d(x_k, y) \big) \mid y \in S \}\!\}.$$

As before, we let $b = \frac{1}{|S|} \sum_{y \in S} y$ denote the barycenter of $S$. For a finite set $G \subset \mathbb{R}^d$, we denote by $\mathsf{LinearSpan}(G)$ the linear space spanned by $G$, and by $\mathsf{AffineSpan}(G)$ the corresponding affine one. Their respective dimensions will be denoted by $\mathsf{LinearDim}(G)$ and $\mathsf{AffineDim}(G)$.

As for the case $d = 2$, we start by distilling the Initialization Data required for the reconstruction, which is described relative to the barycenter $b$ of $S$. For convenience, we have decided not to assume at this stage that $S$ has been translated first to put $b$ at the origin, as we did for the sake of the exposition in the case $d = 2$. This is now the task of the isometry we apply to $S$ when choosing locations for its points, which we now completely relegate to the reconstruction phase.

**Definition 1.** Let $S \in \mathbb{R}^d$ be a finite set and let $b$ be its barycenter. A $d$-tuple $\mathbf{x} = (x_1, \ldots, x_d) \in S^d$ satisfies the ***cone condition*** if

- $\mathsf{AffineDim}(b, x_1, \ldots, x_d) = \mathsf{AffineDim}(S)$;

- if $\mathsf{AffineDim}(S) = d$, then there is no $x \in S$ such that $x - b$ belongs to the interior of $\mathsf{Cone}(x_1 - b, \ldots, x_d - b)$.

**Definition 2.** For a tuple $\mathbf{x} = (x_1, \ldots, x_d) \in S^d$, we define its ***enhanced profile*** as

$$EP(x_1, \ldots, x_d) = (A, M_1, \ldots, M_d),$$

where $A$ is the distance matrix of the tuple $(b, x_1, \ldots, x_d)$ and $M_i = D_{\mathbf{x}[b/i]}$ is the distance profile of the tuple $(x_1, \ldots, x_{i-1}, b, x_{i+1}, \ldots, x_d)$ with respect to $S$.

**Definition 3.** Let $S \in \mathbb{R}^d$ be a finite set and let $b$ be its barycenter. An ***initialization data*** for $S$ is a tuple $(A, M_1, \ldots, M_d)$ such that $(A, M_1, \ldots, M_d) = EP(x_1, \ldots, x_d)$ for some $d$-tuple $\mathbf{x} = (x_1, \ldots, x_d) \in S^d$ satisfying the cone condition.

*Remark* 4.1. The interested reader can verify that the Initialization Data condition extends the definition for $d = 2$. Note that the first bullet of the Cone Condition is automatically verified for $d = 2$, but is nontrivial for $d > 2$.

The fact that an initialization data $(A, M_1, \ldots, M_d)$ can be recovered from $\mathcal{M}_{d-1}(S)$ is ensured by the following proposition.

**Proposition 4.2** (Initialization Lemma). *Take any $S \subseteq \mathbb{R}^d$. Then, knowing the multiset $\{\!\{ \chi^{(3)}_{d-1, S}(\mathbf{s}) \mid \mathbf{s} \in S^{d-1} \}\!\}$, one can determine an initialization data for $S$.*

We now proceed with the reconstruction phase.

## 4.1 Reconstruction Algorithm

Assume an Initialization Data $(A, M_1, \ldots, M_d)$ for a finite $S \subset \mathbb{R}^d$ is given. Our first task is to choose, up to isometry, positions for the points in the $(d+1)$-tuple $(b, x_1, \ldots, x_d)$ corresponding to the matrix $A$. We use the following classical lemma, whose proof is given e.g. in [4, Sec. 2.2.1].

**Lemma 4.3** (Anchor Lemma). *If $(u_1, \ldots, u_k) \in \mathbb{R}^d$ and $(v_1, \ldots, v_k) \in \mathbb{R}^d$ have the same distance matrix, then there exists an isometry $f \colon \mathbb{R}^d \to \mathbb{R}^d$ such that $f(u_i) = v_i$ for all $i = 1, \ldots, k$.*

As for $d = 2$, we put the barycenter of the cloud at the origin. Then, we simply apply the Anchor Lemma to any collection of points $z_1, \ldots, z_d \in \mathbb{R}^d$ such that our given $A$ is also the distance matrix of the tuple $(0, z_1, \ldots, z_d)$. The Lemma then gives us an isometry $f \colon \mathbb{R}^d \to \mathbb{R}^d$ such that $f(b) = 0, f(x_1) = z_1, \ldots, f(x_d) = z_d$. As distance profiles are invariant under isometries, our given $M_i$ is also the distance profile of the tuple $(z_1, \ldots, z_{i-1}, 0, z_{i+1}, \ldots, z_d)$ w.r.t. $f(S)$. Our task now is, from $M_1, \ldots, M_d$, to uniquely determine the locations of all points in $f(S)$. This would give us $S$ up to an isometry. Since now we have locations for $(z_1, \ldots, z_d)$, we can in fact compute:

$$\mathsf{AffineDim}(S) = \mathsf{AffineDim}(b, x_1, \ldots, x_d) = \mathsf{AffineDim}(0, z_1, \ldots, z_d) = \mathsf{LinearDim}(z_1, \ldots, z_d),$$

where the first equality is guaranteed by the cone condition. The reconstruction algorithm depends on whether $\mathsf{AffineDim}(S) = d$ or not.

Consider first the case when $\mathsf{AffineDim}(S) = \mathsf{LinearDim}(z_1, \ldots, z_d) < d$. Then there exists $i \in \{1, \ldots, d\}$ such that $\mathsf{LinearDim}(z_1, \ldots, z_d) = \mathsf{LinearDim}(z_1, \ldots, z_{i-1}, z_{i+1}, \ldots, z_d)$. This means that $f(S) \subset \mathsf{AffineSpan}(z_1, \ldots, z_{i-1}, 0, z_{i+1}, \ldots, z_d)$, since otherwise $f(S)$ would have larger affine dimension. We claim that, in this case, from $M_i$ we can restore the location of all points in $f(S)$. Indeed, from $M_i$ we know, for each $z \in f(S)$, a tuple with the distances from $z$ to $z_1, \ldots, z_{i-1}, 0, z_{i+1}, \ldots, z_d$. As the next lemma shows, this information is enough to uniquely determine the location of $z$.

**Lemma 4.4.** *Take any $x_1, \ldots, x_m \in \mathbb{R}^d$. Assume that $a, b \in \mathsf{AffineSpan}(x_1, \ldots, x_m)$ are such that $d(a, x_i) = d(b, x_i)$ for all $i = 1, \ldots, m$. Then $a = b$.*

It remains to reconstruct $f(S)$ when $\mathsf{AffineDim}(S) = d$, in which case our pivot points $z_1, \ldots, z_d$ are linearly independent. Recall that $M_i$ is the distance profile of the tuple $(z_1, \ldots, z_{i-1}, 0, z_{i+1}, \ldots, z_d)$ w.r.t. $f(S)$. Moreover, since no $x$ in $S$ is such that $x - b$ lies in the interior of $\mathsf{Cone}(x_1 - b, \ldots, x_d - b)$, we know that $f(S)$ must be disjoint from the interior of $\mathsf{Cone}(z_1, \ldots, z_d)$. As the next proposition shows, this information is enough to reconstruct $f(S)$ in this case as well, which finishes the proof of Theorem 2.1 for $d > 2$.

**Proposition 4.5** (The Reconstruction Lemma). *Assume that $z_1, \ldots, z_d \in \mathbb{R}^d$ are linearly independent. Let $T \subseteq \mathbb{R}^d$ be finite and disjoint from the interior of $\mathsf{Cone}(z_1, \ldots, z_d)$. If, for every $i = 1, \ldots, d$, we are given $z_i$ and also the distance profile of the tuple $(z_1, \ldots, z_{i-1}, 0, z_{i+1}, \ldots, z_d)$ w.r.t. $T$, then we can uniquely determine $T$.*

# 5 On the distinguishing power of one iteration of $d$-WL

In this section, we discuss a somewhat different strategy to reconstruct $S$. It is clear that if for a point $z \in \mathbb{R}^d$ we are given the distances from it to $d + 1$ points in general position with known coordinates, then the position of $z$ is uniquely determined (see e.g. Lemma 4.4). Since $d$-WL colors $d$-tuples of points in $S$, a natural strategy to recover $S$ is to use the barycenter as an additional point. By Lemma 3.1, we know that distances to the barycenter from points of $S$ can be obtained after one iteration of $d$-WL. However, the information that allows us to match $d(z, b)$ to the distances from this $z$ to a $d$-tuple, is readily available only after two iterations of $d$-WL. It follows that this simple strategy can be used to directly reconstruct $S$ from the second iteration of $d$-WL. We remark that this strategy is similar to the one used in [9] to uniquely determine $S$ up to isometries when the coloring we are initially given corresponds to certain Gram Matrices for $d$-tuples of points. Essentially, after one interaction of $d$-WL over this initial data, we obtain enough information to directly determine the location of each $z$ relative to a collection of $d + 1$ points. In fact, it is not hard to show that from the first iteration of $d$-WL, applied to the distance graph of $S$, one can compute these Gram Matrices, thus providing an alternative proof that two iterations suffice for distinguishing geometric graphs.

We show instead that only one iteration suffices. Our approach differs and depends on certain geometric principles that allow us to simplify the problem by conducting an exhaustive search across an exponentially large range of possibilities.

**Theorem 5.1.** *For any $d \geq 1$ and for any finite set $S \subseteq \mathbb{R}^d$, knowing the multiset $\{\!\{\chi_{d,S}^{(1)}(\mathbf{s}) | \mathbf{s} \in S^d\}\!\}$, one can determine $S$ up to an isometry.*

# 6 Final remarks

**Open problems.** An interesting open question about our work is what is the minimum number of rounds needed for the $(d-1)$-WL test to be complete with respect to point clouds in $\mathbb{R}^d$. Our result shows that three rounds suffice, but we do not know the completeness status of the test when only one or two rounds are allowed. Another open problem is the completeness status of the $(d-2)$-WL test for $d$-dimensional point clouds when $d > 3$.

**Limitations.** A consequence of our main result and is that distance-based $d$-MPGNNs possess sufficient expressive capability to learn $d$-dimensional point clouds up to isometry. However, the computational complexity of implementing $d$-MPGNNs is a major concern, as it involves $O(n^d)$ operations per iteration, where $n$ denotes the number of nodes in the graph. This computational cost can quickly become unmanageable, even for relatively small values of $d$, such as $d = 3$. It remains to be studied which kind of optimizations on higher-order GNNs can be implemented for improved performance without much sacrifice on their expressive power.

# 7 Acknowledgements

Barceló and Kozachinskiy are funded by ANID–Millennium Science Initiative Program - ICN17002. All authors are funded by the National Center for Artificial Intelligence CENIA FB210017, Basal ANID. Delle Rose is funded by ANID Fondecyt Postdoctorado 3230263. Petrache is funded by ANID Fondecyt Regular 1210462.

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

# A Appendix

## A.1 Proofs for Initialization

Here we provide the proof of Proposition 4.2. We will need the following extension of the Barycenter Lemma.

**Lemma A.1.** *Let $b$ be the barycenter of $S \subset \mathbb{R}^d, d > 2$. For any $\mathbf{x} = (x_1, \ldots, x_{d-1}) \in S^{d-1}$, knowing $\chi^{(1)}_{d-1,S}(\mathbf{x})$ and the multiset $\{\!\{ \chi^{(1)}_{d-1,S}(\mathbf{y}) \mid \mathbf{y} \in S^{d-1} \}\!\}$, one can determine the tuple of distances $(d(x_1, b), \ldots, d(x_{d-1}, b))$.*

*Proof.* Let $D_x$ be as in Lemma 3.1. We claim that we can determine $(D_{x_1}, \ldots, D_{x_{d-1}})$ and $\{\!\{ D_y \mid y \in S \}\!\}$ from the information given in the lemma statement. By Lemma 3.1, this allows to determine $(d(x_1, b), \ldots, d(x_{d-1}, b))$.

We have

$$\chi^{(1)}_{d-1,S}(\mathbf{x}) = \left( \chi^{(0)}_{d-1,S}(\mathbf{x}), \{\!\{ \left( \chi^{(0)}_{d-1,S}(\mathbf{x}[y/1]), \ldots, \chi^{(0)}_{d-1,S}(\mathbf{x}[y/(d-1)]) \right) \mid y \in S \}\!\} \right).$$

By definition, from $\chi^{(0)}_{d-1,S}(\mathbf{x}[y/1])$ one can determine the tuple of distances $(d(x_2, y), \ldots, d(x_{d-1}, y))$. Hence, from the multiset $\{\!\{ \chi^{(0)}_{d-1,S}(\mathbf{x}[y/1]) \mid y \in S \}\!\}$ one can determine $(D_{x_2}, \ldots, D_{x_{d-1}})$. In turn, $D_{x_1}$ can be determined from, say, $\{\!\{ \chi^{(0)}_{d-1,S}(\mathbf{x}[y/2]) \mid y \in S \}\!\}$. We have just shown that $\chi^{(1)}_{d-1,S}(\mathbf{x})$ uniquely determines $D_{x_1}$. Hence, from the multiset $\{\!\{ \chi^{(1)}_{d-1,S}(\mathbf{y}) \mid \mathbf{y} \in S^{d-1} \}\!\}$, one can compute the multiset $\{\!\{ D_{y_1} \mid \mathbf{y} = (y_1, \ldots, y_{d-1}) \in S^{d-1} \}\!\}$. But this multiset coincides with the multiset $\{\!\{ D_y \mid y \in S \}\!\}$ except that all multiplicities are $|S|^{d-2}$ times larger. $\qquad\square$

Now, we show that after the second iteration, we can restore the distance profile of $(b, x_1, \ldots, x_{d-1})$ for all $(x_1, \ldots, x_{d-1}) \in S^{d-1}$.

**Lemma A.2.** *For any $\mathbf{x} = (x_1, \ldots, x_{d-1}) \in S^{d-1}$, knowing $\chi^{(2)}_{d-1,S}(\mathbf{x})$ and the multiset $\{\!\{ \chi^{(1)}_{d-1,S}(\mathbf{y}) \mid \mathbf{y} \in S^{d-1} \}\!\}$, one can determine the distance profile of $(b, x_1, \ldots, x_{d-1})$ w.r.t. $S$.*

*Proof.* Since $d > 2$, we have

$$\chi^{(2)}_{d-1,S}(\mathbf{x}) = \left( \chi^{(1)}_{k,S}(\mathbf{x}), \{\!\{ \left( \chi^{(1)}_{d-1,S}(\mathbf{x}[y/1]), \ldots, \chi^{(1)}_{d-1,S}(\mathbf{x}[y/k]) \right) \mid y \in S \}\!\} \right).$$

From the tuple $\left( \chi^{(0)}_{d-1,S}(\mathbf{x}[y/1]), \ldots, \chi^{(0)}_{d-1,S}(\mathbf{x}[y/k]) \right)$ we can restore the tuple of distances

$$(d(y, x_1), \ldots, d(y, x_{d-1})).$$

In turn, by Lemma A.1, from $\chi^{(1)}_{d-1,S}(\mathbf{x}[y/1])$ and $\{\!\{ \chi^{(1)}_{d-1,S}(\mathbf{y}) \mid \mathbf{y} \in S^{d-1} \}\!\}$, we can restore $d(y, b)$. Hence, we can restore the whole distance profile of $(b, x_1, \ldots, x_{d-1})$. $\qquad\square$

Finally, we show that knowing $\chi^{(3)}_{d-1,S}(\mathbf{x})$ for $\mathbf{x} = (x_1, \ldots, x_{d-1})$ and the multiset $\{\!\{ \chi^{(1)}_{d-1,S}(\mathbf{y}) \mid \mathbf{y} \in S^{d-1} \}\!\}$, we can compute

$$\{ EP(x_1, \ldots, x_{d-1}, y) \mid y \in S \}.$$

Since $d > 2$, we have

$$\chi^{(3)}_{d-1,S}(\mathbf{x}) = \left( \chi^{(2)}_{d-1,S}(\mathbf{x}), \{\!\{ \left( \chi^{(2)}_{d-1,S}(\mathbf{x}[y/1]), \ldots, \chi^{(2)}_{d-1,S}(\mathbf{x}[y/k]) \right) \mid y \in S \}\!\} \right).$$

For every $y \in S$, knowing $\chi^{(2)}_{d-1,S}(\mathbf{x})$ and $\left( \chi^{(2)}_{d-1,S}(\mathbf{x}[y/1]), \ldots, \chi^{(2)}_{d-1,S}(\mathbf{x}[y/k]) \right)$, we have to compute $EP(x_1, \ldots, x_{d-1}, y)$, that is, the distance matrix of $(b, x_1, \ldots, x_{d-1}, y)$ and the distance profiles of the tuples

$$(b, x_2, \ldots, x_{d-1}, y), \ldots, (x_1, \ldots, x_{d-2}, b, y), (x_1, \ldots, x_{d-1}, b).$$

Distance profiles can be computed by Lemma A.2. The distances amongst elements of $\{x_1, \ldots, x_{d-1}, y\}$ can be computed by definition from $\chi^{(0)}_{d-1,S}(\mathbf{x})$ and $\left(\chi^{(0)}_{d-1,S}(\mathbf{x}[y/1]), \ldots, \chi^{(0)}_{d-1,S}(\mathbf{x}[y/k])\right)$. Distances to $b$ from these points can be computed by Lemma A.1 from $\chi^{(1)}_{d-1,S}(\mathbf{x})$ and $\left(\chi^{(1)}_{d-1,S}(\mathbf{x}[y/1]), \ldots, \chi^{(1)}_{d-1,S}(\mathbf{x}[y/k])\right)$.

Now that we have the enhanced profiles, we have to select one for which $(x_1, \ldots, x_d)$ satisfies the cone condition. We first observe that, knowing $EP(x_1, \ldots, x_d)$, we can reconstruct $(b, x_1, \ldots, x_d)$ up to an isometry by Lemma 4.3 (because inside $EP(x_1, \ldots, x_d)$ we are given the distance matrix of $(b, x_1, \ldots, x_d)$). This means that from $EP(x_1, \ldots, x_d)$ we can compute any function of $b, x_1, \ldots, x_d$ which is invariant under isometries. In particular, we can compute AffineDim$(b, x_1, \ldots, x_d)$. We will refer to AffineDim$(b, x_1, \ldots, x_d)$ as the dimension of the corresponding enhanced profile.

We show that AffineDim$(S)$ is equal to the maximal dimension of an enhanced profile. Indeed, first notice that AffineDim$(S) = $ AffineDim$(\{b\} \cup S)$ because $b$ is a convex combination of points of $S$. In turn, AffineDim$(\{b\} \cup S)$ is equal to the maximal $k$ for which one can choose $k$ points $x_1, \ldots, x_k \in S$ such that $x_1 - b, \ldots, x_k - b$ are linearly independent. Obviously, $k$ is bounded by the dimension of the space. Hence, there will be an enhanced profile with the same maximal dimension $k$.

If AffineDim$(S) < d$, any enhanced profile with maximal dimension satisfies the initialization requirement, and we are done. Assume therefore that AffineDim$(S) = d$. Our task is to output some $EP(x_1, \ldots, x_d)$ such that

1. AffineDim$(b, x_1, \ldots, x_d) = d$, and
2. there is no $x \in S$ such that $x - b$ belongs to the interior of Cone$(x_1 - b, \ldots, x_d - b)$.

For that, among all $d$-dimensional enhanced profiles, we output one which minimizes the solid angle at the origin, defined as

$$\mathsf{Angle}(x_1 - b, \ldots, x_d - b) = \frac{1}{d}\mathsf{Vol}\left\{x \in \mathsf{Cone}(x_1 - b, \ldots, x_d - b) : \|x\| \le 1\right\} \qquad (3)$$

(the solid angle is invariant under isometries, and hence can be computed from $EP(x_1, \ldots, x_d)$).

We have to show that for all $x \in S$ we have that $x - b$ lies outside the interior of Cone$(x_1 - b, \ldots, x_d - b)$. To prove this, we need an extra lemma. We will say that a cone $C$ is *simple* if $C = $ Cone$(u_1, \ldots, u_d)$, for some linearly independent $u_1, \ldots, u_d$. Observe that if $C = $ Cone$(u_1, \ldots, u_d)$ is a simple cone, then the interior of $C$ is the set

$$int(C) = \{\lambda_1 u_1 + \ldots + \lambda_d u_d \mid \lambda_1, \ldots, \lambda_d \in (0, +\infty)\}.$$

Note also that the boundary of $C$ consists of $d$ faces

$$F_i = C \cap \mathsf{LinearSpan}(u_1, \ldots, u_{i-1}, u_{i+1}, \ldots, u_d), \quad i = 1, \ldots, d.$$

**Lemma A.3.** *Let $C = \mathsf{Cone}(\mathbf{u}) \subseteq \mathbb{R}^d$ for $\mathbf{u} = (u_1, \ldots, u_d)$ be a simple cone and let $y$ belong to the interior of $C$. Then $\mathsf{Cone}(\mathbf{u}[y/1])$ is a simple cone and $\mathsf{Angle}(\mathbf{u}[y/1]) < \mathsf{Angle}(\mathbf{u})$.*

*Proof.* Since $y$ belongs to the interior of Cone$(\mathbf{u})$, we have that

$$y = \lambda_1 u_1 + \ldots + \lambda_d u_d,$$

for some $\lambda_1 > 0, \ldots, \lambda_d > 0$. The fact that $\lambda_1 > 0$ implies that $y, u_2, \ldots, u_d$ are linearly independent, and hence Cone$(\mathbf{u}[y/1])$ is a simple cone. Since $y \in $ Cone$(\mathbf{u})$, we have that Cone$(\mathbf{u}[y/1]) \subseteq $ Cone$(\mathbf{u})$. Thus, to show that Angle$(\mathbf{u}[y/1]) < $ Angle$(\mathbf{u})$, it suffices to show that the volume of

$$\{x \in \mathsf{Cone}(\mathbf{u}) : \|x\| \le 1\} \setminus \{x \in \mathsf{Cone}(\mathbf{u}[y/1]) : \|x\| \le 1\}$$

is positive. We claim that for any point $x = \mu_1 u_1 + \ldots + \mu_d u_d \in $ Cone$(\mathbf{u}[y/1])$ we have $\mu_1 > 0 \implies \mu_2/\mu_1 \ge \lambda_2/\lambda_1$. This is because $x$ can be written as a non-negative linear combination of $y, u_2, \ldots, u_d$. Since $\mu_1 > 0$, the coefficient in front of $y$ in this linear combination must be positive. Now, if the coefficient in front of $u_2$ is 0, then the ratio between $\mu_2$ and $\mu_1$ is exactly as the ratio between $\lambda_2$ and $\lambda_1$, and if the coefficient before $u_2$ is positive, $\mu_2$ can only increase.

This means that no point of the form

$$x = \mu_1 u_1 + \ldots + \mu_d u_d, \qquad 0 < \mu_1, \ \mu_2/\mu_1 < \lambda_2/\lambda_1 \tag{4}$$

belongs to $\mathsf{Cone}(\mathbf{u}[y/1])$. It remains to show that the set of points that satisfy (4) and lie in $\{x \in \mathsf{Cone}(\mathbf{u}) : \|x\| \leq 1\}$ has positive volume.

Indeed, for any $\varepsilon > 0$ and $\delta > 0$, consider a $d$-dimensional parallelepiped:

$$P = \{\mu_1 x_1 + \ldots + \mu_d x_d \mid \mu_1 \in [\varepsilon/2, \varepsilon], \ \mu_2, \ldots, \mu_d \in [\delta/2, \delta]\}.$$

Regardless of $\varepsilon$ and $\delta$, we have that $P$ is a subset of $\mathsf{Cone}(\mathbf{u})$ and its volume is positive. For all sufficiently small $\varepsilon, \delta$, we have that $P$ is a subset of the unit ball $\{x \in \mathbb{R}^d \mid \|x\| \leq 1\}$. In turn, by choosing $\varepsilon$ to be sufficiently big compared to $\delta$, we ensure that all points of $P$ satisfy (4). $\qquad\square$

Now we can finish the proof of Proposition 4.2. Let $\mathbf{x} \in S^d$ minimize $\mathsf{Angle}(x_1 - b, \ldots, x_d - b)$ amongst $\mathbf{x} \in S^d$ such that $\mathsf{AffineDim}(b, x_1, \ldots, x_d) = d$. Assume that $\mathbf{x}$ does not satisfy the cone condition, and let $x \in S$ be such that $x - b \in Int(\mathsf{Cone}(x_1 - b, \ldots, x_d - b))$ holds. Then, by Lemma A.3, $\mathsf{Angle}(x - b, x_2 - b \ldots, x_d - b)$ is strictly smaller than $\mathsf{Angle}(x_1 - b, x_2 - b \ldots, x_d - b)$. As $\mathsf{Cone}(x - b, x_2 - b \ldots, x_d - b)$ is a simple cone, $x - b, x_2 - b, \ldots, x_d - b$ are linearly independent, and thus $\mathsf{AffineDim}(b, x, x_2, \ldots, x_d) = d$, contradicting the minimality hypothesis on $\mathbf{x}$, as desired.

## A.2 Proofs for Reconstruction

*Proof of Lemma 4.4.* We claim that for every $i = 2, \ldots, m$ we have $\langle a - x_1, x_i - x_1 \rangle = \langle b - x_1, x_i - x_1 \rangle$. This is because $a - x_1$ and $b - x_1$ have the same distance to $x_i - x_1$ (which is equal to $d(a, x_i) = d(b, x_i)$) and, moreover, the norms of $a - x_1$ and $b - x_1$ coincide (and are equal to $d(a, x_1) = d(b, x_1)$). Hence, both $a - x_1$ and $b - x_1$ are solutions to the following linear system of equations:

$$\langle x, x_i - x_1 \rangle = \langle a - x_1, x_i - x_1 \rangle, \qquad i = 2, \ldots, m.$$

This system has at most 1 solution over $x \in \mathsf{LinearSpan}(x_2 - x_1, \ldots, x_m - x_1)$. Moreover, $a - x_1$ and $b - x_1$ are both from $\mathsf{LinearSpan}(x_2 - x_1, \ldots, x_m - x_1)$ because $a, b \in \mathsf{AffineSpan}(x_1, \ldots, x_m)$. Hence, $a - x_1 = b - x_1$, and $a = b$. $\qquad\square$

*Proof of Proposition 4.5.* Let $P_i = \mathsf{AffineSpan}(z_1, \ldots, z_{i-1}, 0, z_{i+1}, \ldots, z_d)$. As the following lemma shows, knowing the distances from $s \in T$ to $z_1, \ldots, z_{i-1}, 0, z_{i+1}, z_d$, we can determine the position of $s$ uniquely up to the reflection through $P_i$.

**Lemma A.4** (The Symmetric Lemma). *Let $x_1, \ldots, x_m \in \mathbb{R}^d$ be such that $\mathsf{AffineSpan}(x_1, \ldots, x_m)$ has dimension $d - 1$. Assume that $a, b \in \mathbb{R}^d$ are such that $d(a, x_i) = d(b, x_i)$ for all $i = 1, \ldots, m$. Then either $a = b$ or $a$ and $b$ are symmetric w.r.t. $\mathsf{AffineSpan}(x_1, \ldots, x_m)$.*

*Proof.* As in the proof of Lemma 4.4, we have that $\langle a - x_1, x_i - x_1 \rangle = \langle b - x_1, x_i - x_1 \rangle$ for every $i = 2, \ldots, m$. Consider orthogonal projections of $a - x_1$ and $b - x_1$ to $\mathsf{LinearSpan}(x_2 - x_1, \ldots, x_m - x_1)$. Both these projections are solutions to the system

$$\langle x, x_i - x_1 \rangle = \langle a - x_1, x_i - x_1 \rangle, \qquad i = 2, \ldots, m.$$

This system has at most one solution over $x \in \mathsf{LinearSpan}(x_2 - x_1, \ldots, x_m - x_1)$. Hence, projections of $a - x_1$ and $b - x_1$ coincide. We also have that $\|a - x_1\| = \|b - x_1\|$, which implies that $a - x_1$ and $b - x_1$ have the same distance to $\mathsf{LinearSpan}(x_2 - x_1, \ldots, x_m - x_1)$. Since the dimension of $\mathsf{LinearSpan}(x_2 - x_1, \ldots, x_m - x_1)$ is $d - 1$, we get that either $a - x_1 = b - x_1$ or they can be obtained from each other by the reflection through $\mathsf{LinearSpan}(x_2 - x_1, \ldots, x_m - x_1)$. After translating everything by $x_1$, we obtain the claim of the lemma. $\qquad\square$

In fact, if $s$ belongs to $P_i$, then there is just one possibility for $s$. Thus, we can restore all the points in $T$ that belong to the union $\bigcup_{i=1}^d P_i$. Let us remove these points from $T$ and update distance profiles by deleting the tuples of distances that correspond to the points that we have removed.

From now on, we may assume that $T$ is disjoint from $\bigcup_{i=1}^d P_i$. Hence, $T$ is also disjoint from the boundary of $C = \mathsf{Cone}(x_1, \ldots, x_d)$, not only from its interior (every face of this cone lies on $P_i$ for some $i$).

For $x \in \mathbb{R}^d$, we define $\rho(x) = \min_{i=1,\ldots,d} \mathsf{dist}(x, P_i)$. Since $T$ is disjoint from $\bigcup_{i=1}^d P_i$, we have that $\rho(s) > 0$ for every $s \in T$. Moreover, from, say, the distance profile of $(0, z_2, \ldots, z_d)$, we can compute some $\varepsilon > 0$ such that $\rho(s) \geq \varepsilon$ for all $s \in T$. Indeed, recall that from the distance profile of $(0, z_2, \ldots, z_d)$, we get at most 2 potential positions for each point of $T$. This gives us a finite set $T'$ (at most 2 times larger than $T$) which is a superset of $T$. Moreover, as $T$ is disjoint from $\bigcup_{i=1}^d P_i$, we have that $T' \setminus \bigcup_{i=1}^d P_i \supseteq T$ Thus, we can define $\varepsilon$ as the minimum of $\rho(x)$ over $T' \setminus \bigcup_{i=1}^d P_i \supseteq T$.

We conclude that $T$ is disjoint from

$$A_0 = C \cup \{x \in \mathbb{R}^d \mid \rho(x) < \varepsilon\}$$

(moreover, the set $A_0$ is known to us).

Our reconstruction procedure starts as follows. We go through all distance profiles, and through all tuples of distances in them. Each tuple gives 2 candidates for a point in $T$ (that can be obtained from each other by the reflection through $P_i$). If one of the candidates lies in $A_0$, we know that we should take the other candidate. In this way, we may possibly uniquely determine some points in $T$. If so, we remove them from $T$ and update our distance profiles.

Which points of $T$ will be found in this way? Those that, for some $i$, fall into $A_0$ under the reflection through $P_i$. Indeed, these are precisely the points that give 2 candidates (when we go through the $i$th distance profile) one of which is in $A_0$. In other words, we will determine all the points that lie in $\bigcup_{i=1}^d \mathsf{Refl}_i(A_0)$, where $\mathsf{Refl}_i$ denotes the reflection through $P_i$. After we remove these points, we know that the remaining $T$ is disjoint from $A_1 = A_0 \cup \bigcup_{i=1}^d \mathsf{Refl}_i(A_0)$.

We then continue in exactly the same way, but with $A_1$ instead of $A_0$, and then with $A_2 = A_1 \cup \bigcup_{i=1}^d \mathsf{Refl}_i(A_1)$, and so on. It remains to show that all the points of $T$ will be recovered in this way. In other words, we have to argue that each point of $T$ belongs to some $A_i$, where

$$A_0 = C \cup \{x \in \mathbb{R}^d \mid \rho(x) < \varepsilon\}, \qquad A_{i+1} = A_i \cup \bigcup_{i=1}^d \mathsf{Refl}_i(A_i).$$

We will show this not only for points in $T$ but for all points in $\mathbb{R}^d$. Equivalently, we have to show that for every $x \in \mathbb{R}^d$ there exists a finite sequence of reflections $\tau_1, \ldots, \tau_k \in \{\mathsf{Refl}_1, \ldots, \mathsf{Refl}_d\}$ which brings $x$ inside $A_0$, that is, $\tau_k \circ \ldots \circ \tau_1(x) \in A_0$.

We construct this sequence of reflections as follows. Let $x$ be outside $A_0$. In particular, $x$ is outside the cone $C = \mathsf{Cone}(z_1, \ldots, z_d)$. Then there exists a face of this cone such that $C$ is from one side of this face and $x$ is from the other side. Assume that this face belongs to the hyperplane $P_i$. We then reflect $x$ through $P_i$, and repeat this operation until we get inside $A_0$. We next show that the above process stops within a finite number of steps. For that, we introduce the quantity $\gamma(x) = \langle x, z_1 \rangle + \ldots + \langle x, z_d \rangle$. We claim that with each step, $\gamma(x)$ increases by at least $c \cdot \varepsilon$, where

$$c = 2 \min_{1 \leq i \leq d} \mathsf{dist}(z_i, P_i).$$

Note that $c > 0$ because, for every $i = 1, \ldots, d$, we have that $z_i \notin P_i$, by the linear independency of $z_1, \ldots, z_d$. Also observe that $\gamma(x) \leq |x| \sum_i |z_i|$ and reflections across the subspaces $P_i$ do not change $|x|$. Hence, $\gamma(x)$ cannot increase infinitely many times by some fixed positive amount.

It remains to show that $\gamma(x)$ increases by at least $c \cdot \varepsilon$ at each step, as claimed. Note that reflection of $x$ across some $P_i$ does not change the scalar product of $x$ with those vectors among $z_1, \ldots, z_d$ that belong to $P_i$. The only scalar product that changes is $\langle x, z_i \rangle$, and the only direction which contributes to the change is the one orthogonal to $P_i$. Before the reflection, the contribution of this direction to the scalar product was $-d(x, P_i) \cdot d(z_i, P_i)$ (remember that $x$ and $z_i$ were from different sides of $P_i$ because $z_i \in C$). After the reflection, the contribution is the same, but with a positive sign. Thus, the scalar product increases by $2d(x, P_i) \cdot d(z_i, P_i)$. Now, we have $d(z_i, P_i) \geq c/2$ by definition of $c$ and $d(x, P_i) \geq \varepsilon$ if $x$ is not yet in $A_0$. $\qquad\square$

# B  Proofs for Theorem 5.1

*Proof of Theorem 5.1.* From $\chi_{d,S}^{(1)}(\mathbf{s})$, we can determine the tuple $\mathbf{s} = (s_1, \ldots, s_d)$ up to an isometry, since $\chi_{d,S}^{(1)}(\mathbf{s})$ gives us $\chi_{d,S}^{(0)}(\mathbf{s})$, which is the distance matrix of $\mathbf{s}$. In order to determine $S$, we consider two cases:

**Case 1**: *For all* $\mathbf{s} \in S^d$ *it holds* $\mathsf{AffineDim}(\mathbf{s}) < d-1$. Then take $\chi_{d,S}^{(1)}(\mathbf{s})$ with maximal $\mathsf{AffineDim}(\mathbf{s})$, and fix locations of points from $\mathbf{s}$ compatible with the distance specification, according to Lemma 4.3. All points of $S$ belong to $\mathsf{AffineSpan}(\mathbf{s})$, otherwise we could increase $\mathsf{AffineDim}(\mathbf{s})$. Indeed, since $\mathsf{AffineDim}(\mathbf{s}) < d - 1$, we could throw away one of the points from the tuple without decreasing the dimension and replace it with a point outside $\mathsf{AffineSpan}(\mathbf{s})$. We now can reconstruct the rest of $S$ uniquely up to an isometry. Indeed, in $\chi_{d,S}^{(1)}(\mathbf{s})$ we are given the multiset of $d$-tuples of distances to $\mathbf{s} = (s_1, \ldots, s_d)$ from the points of $S$, and it remains to use Lemma 4.4.

**Case 2**: *There are tuples with* $\mathsf{AffineSpan}(\mathbf{s}) = d - 1$. We first observe that from the multiset $\{\!\{ \chi_{d,S}^{(0)}(\mathbf{s}) \mid \mathbf{s} \in S^d \}\!\}$ we can compute the pairwise sum of distances between the points in $S$, i.e.,

$$D_S = \sum_{x \in S} \sum_{y \in S} d(x, y).$$

Indeed, from $\chi_{d,S}^{(0)}((s_1, \ldots, s_d))$, we determine $d(s_1, s_2)$. Hence, we can compute the sum:

$$\sum_{(s_1, \ldots, s_d) \in S^d} d(s_1, s_2) = D_S \cdot |S|^{d-2}.$$

In our reconstruction of $S$, we go through all $\chi_{d,S}^{(1)}(\mathbf{s})$ with $\mathsf{AffineDim}(\mathbf{s}) = d - 1$. For each of them, we fix positions of the points of the tuple $\mathbf{s}$ in any way that agrees with the distance matrix of this tuple. As before, $\chi_{d,S}^{(1)}(\mathbf{s})$ gives us the multiset of $d$-tuples of distances to $\mathbf{s}$ from the points of $S$. We call "candidates for $S$ given $\mathbf{s}$" the set of point clouds $S'$ which have one point associated with each such $d$-tuple of distances, and realizing these distances to points in $\mathbf{s}$. We aim to find $\mathbf{s}$ for which, exactly one of these candidates can be isometric to $S$. We start with the following lemma:

**Lemma B.1.** *For any finite set $S \subseteq \mathbb{R}^d$ with $\mathsf{AffineDim}(S) \geq d - 1$ there exist $x_1, \ldots, x_d \in S$ with $\mathsf{AffineDim}(x_1, \ldots, x_d) = d - 1$ such that all points of $S$ belong to the same half-space with respect to the hyperplane $\mathsf{AffineSpan}(x_1, \ldots, x_d)$.*

*Proof.* The general idea of the proof is the following. If $\mathsf{AffineDim}(S) = d - 1$ then the extreme points of the convex hull of $S$ contain an affinely independent set of cardinality $d$, which then gives the desired $\mathbf{s}$. The half-space condition in the lemma is automatically verified in this case. If $\mathsf{AffineDim}(S) = d$ then to find $\mathbf{s}$ we can proceed by moving a $(d-1)$-plane from infinity towards $S$ until it touches $S$ in at least one point, then iteratively we rotate the plane around the subspace containing the already touched points of $S$, until a new point in $S$ prohibits to continue that rotation. We stop within at most $d$ iterations, when no further rotation is allowed, in which case the plane has an affinely independent subset in common with $S$.

Formally, we need to find a hyperplane $H$ such that, first, all points of $S$ belong to the same half-space w.r.t. $H$, and second, $\mathsf{AffineDim}(H \cap S) = d - 1$.

To start, we need to find a hyperplane $H$ such that, first, all points of $S$ belong to the same half-space w.r.t. $H$, and second, $H \cap S \neq \varnothing$. For instance, take any non-zero vector $\alpha \in \mathbb{R}^d$, consider $m = \max_{x \in S} \langle \alpha, x \rangle$ and define $H$ by the equation $\langle \alpha, x \rangle = m$. Now, take any $x_1 \in H \cap S$. After translating $S$ by $-x_1$, we may assume that $x_1 = 0$.

Now, among all hyperplanes $H$ that contain $x_1 = 0$ and satisfy the condition that all points of $S$ lie in the same half-space w.r.t. $H$, we take one that contains most points of $S$. We claim that $\mathsf{AffineDim}(H \cap S) = d - 1$ for this $H$. Assume for contradiction that $\mathsf{AffineDim}(H \cap S) < d - 1$. Define $U = \mathsf{AffineSpan}(H \cap S)$. Since $H \cap S$ contains $x_1 = 0$, we have that $U \subseteq H$ is a linear subspace, and its dimension is less than $d - 1$. Hence, since $\mathsf{AffineDim}(S) \geq d - 1$, there exists $x_2 \in S \setminus U$. Note that $x_2 \notin H$ because otherwise $x_2$ belongs to $H \cap S \subseteq U$.

Let $\alpha$ be the normal vector to $H$. Since all points of $S$ lie in the same half-space w.r.t. $H$, w.l.o.g. we may assume that $\langle \alpha, s \rangle \geq 0$ for all $s \in S$. In particular, $\langle \alpha, x_2 \rangle > 0$ because $x_2 \notin H$.

Let $U^\perp$ denote the orthogonal complement to $U$. Since $\alpha$ is the normal vector to $H \supseteq U$, we have that $\alpha \in U^\perp$. We need to find some $\beta \in U^\perp$ which is not a multiple of $\alpha$ but satisfies $\langle \beta, x_2 \rangle > 0$. Indeed, the dimension of $U$ is at most $d - 2$, and hence the dimension of $U^\perp$ is at least 2. Now, since $\langle \alpha, x_2 \rangle > 0$, we can take any $\beta \in U^\perp$ which is sufficiently close to $\alpha$.

For any $\lambda \geq 0$, let $H_\lambda$ be the hyperplane, defined by $\langle \alpha - \lambda\beta, x \rangle = 0$ (this is a hyperplane and not the whole space because $\beta$ is not a multiple of $\alpha$). We claim that for some $\lambda > 0$, we have that $H_\lambda$ has more points of $S$ than $H$ while still all points of $S$ lie in the same half-space w.r.t. $H_\lambda$. This would be a contradiction.

Indeed, define $S_\beta = \{s \in S \mid \langle s, \beta \rangle > 0\}$. Note that $S_\beta$, by definition of $\beta$, contains $x_2$ and hence is non-empty. Moreover, $S_\beta$ is disjoint from $H \cap S$. This is because $H \cap S \subseteq U$ and $\beta \in U^\perp$.

Define
$$\lambda = \min_{s \in S_\beta} \frac{\langle \alpha, s \rangle}{\langle \beta, s \rangle}$$

First, $H_\lambda \supseteq U \supseteq H \cap S$ because $\alpha - \lambda\beta \in U^\perp$. Moreover, $H_\lambda$ contains at least one point of $S$ which is not in $H$. Namely, it $H_\lambda$ contains any $s \in S_\beta$, establishing the minimum in the definition of $\lambda$ (and recall that $S_\beta$ is disjoint from $H \cap S$). Indeed, for this $s$ we have $\lambda = \frac{\langle \alpha, s \rangle}{\langle \beta, s \rangle}$. Hence, $\langle \alpha, s \rangle - \lambda \langle \beta, s \rangle = 0 = \langle \alpha - \lambda b, s \rangle \implies s \in H_\lambda$.

It remains to show that all points of $S$ lie in the same half-space w.r.t. $H_\lambda$. More specifically, we will show that $\langle \alpha - \lambda\beta, s \rangle \geq 0$ for all $s \in S$. First, assume that $\langle s, \beta \rangle = 0$. Then $\langle \alpha - \lambda\beta, s \rangle = \langle \alpha, s \rangle \geq 0$ because all points of $S$ lie in the "non-negative" half-space w.r.t. $\alpha$. Second, assume that $\langle s, \beta \rangle > 0$. Then $s \in S_\beta$. Hence, by definition of $\lambda$, we have $\lambda \leq \frac{\langle \alpha, s \rangle}{\langle \beta, s \rangle}$. This means that $\langle \alpha - \lambda\beta, s \rangle = \langle \alpha, s \rangle - \lambda \langle \beta, s \rangle \geq 0$, as required. $\square$

Next, consider the following simple geometric observation:

**Lemma B.2.** *Let $P \in \mathbb{R}^d$ be a hyperplane and consider two points $a, b \in \mathbb{R}^d \setminus P$ that lie in same half-space w.r.t. $P$. Let $a', b'$ be the reflections of $a, b$ through $P$. Then $d(a', b') = d(a, b) < d(a, b')$.*

*Proof.* It suffices to restrict to the plane $\mathsf{AffineSpan}(\{a, b, a'\})$, and thus we take $d = 2$ and up to isometry we may fix $P$ to be the $x$-axis, $a = (0, y)$, $b = (x, y')$, $b' = (x, -y')$, with $y, y' > 0$. Then it follows that $d(a, b)^2 = x^2 + (y - y')^2 < x^2 + (y + y')^2 = d(a, b')^2$. As reflections are isometries, $d(a, b) = d(a', b')$. $\square$

**Claim:** If $\mathbf{s}$ is as in Lemma B.1, then we have the following:

- Exactly one of the candidates for $S$ given $\mathbf{s}$, up to reflection across $\mathsf{AffineSpan}(\mathbf{s})$, is completely contained in one of the half-spaces determined by $\mathsf{AffineSpan}(\mathbf{s})$.

- A candidate $S'$ as in the previous point is the only one of the candidates for $S$ given $\mathbf{s}$, up to reflection across $\mathsf{AffineSpan}(\mathbf{s})$, for which $D_{S'} = D_S$.

To prove the first item, we use only the property that $\mathsf{AffineDim}(\mathbf{s}) = d - 1$, with which by Lemma A.4, each point of a candidate for $S$ given $\mathbf{s}$, has either two possible locations (related by a reflection across $\mathsf{AffineSpan}(\mathbf{s})$) or a single possible location if it belongs to $\mathsf{AffineSpan}(\mathbf{s})$. For the second item, let $S'$ be as above and let $S''$ be a candidate for $S$ given $\mathbf{s}$, which is not completely contained in one of the halfspaces determined by $\mathsf{AffineSpan}(\mathbf{s})$. We now consider each term $d(x', y')$ in the sum defining $D_{S'}$, comparing the corresponding term $d(x'', y'')$ from $D_{S''}$, where $x'' = x'$ or is a reflection across $\mathbf{s}$ of $x'$ and similarly for $y''$ and $y'$. By Lemma B.2, either $d(x'', y'') = d(x', y')$ in case $x'', y''$ are in the same half-space determined by $\mathsf{AffineSpan}(\mathbf{s})$, or $d(x'', y'') > d(x', y')$ otherwise. Summing all terms, by the property of $S', S''$ we find $D_{S'} < D_{S''}$. By the same reasoning with $S$ instead of $S''$, since we are assuming that $\mathbf{s}$ satisfies Lemma B.2, we have $D_S = D_{S'}$, completing the proof of the second item and of the claim.

The reconstruction of $S$ in Case 2, can therefore be done as follows: we run through all $\mathbf{s}$ such that $\mathsf{AffineDim}(\mathbf{s}) = d - 1$, and for each such $\mathbf{s}$ we construct all candidates $\widetilde{S}$ for $S$ given $\mathbf{s}$, and calculate $D_{\widetilde{S}}$ for each of them. Lemma B.1 guarantees that we run into some $\mathbf{s}$ for which, up to isometry, only one such $\widetilde{S}$ realizes $D_{\widetilde{S}} = D_S$. This is our unique reconstruction of $S$. $\square$

