# OpenReview forum: "Three Iterations of (d − 1)-WL Test Distinguish Non Isometric Clouds of d-dimensional Points"
_NeurIPS.cc/2023/Conference — NeurIPS 2023 poster_

### Official Review · Reviewer_yTgF · 2023-07-02

**Soundness:** 3 good
**Presentation:** 2 fair
**Contribution:** 2 fair
**Rating:** 5
**Confidence:** 3

**Summary:**

This paper studies the completeness of $l$-WL test for Euclidean point sets. It shows an algorithm that certifies two Euclidean point sets of dimension $d$ are isometric using $(d-1)$-WL test, where only three iterations suffice. The results extend to $d$-WL test which only requires one iteration.

The punchline is to study "how pair-wise distances determine the identity of all points", which is well-developed for Euclidean scenarios. Taking the plane as an example, the proposed method shows that storing single-source all-pair distances of 2 points and the norm of the rest points suffice to uniquely reconstruct the whole point set. Such data can be obtained through several rounds of WL-test, which completes the claim that WL-test is able to distinguish the isometry of Euclidean point sets.

**Strengths:**

- The problem is well-motivated, most of the paper is written clearly.
- The results characterize well on three parameters involved for isometry-test of Euclidean point sets: the data dimension $d$, the WL-test dimension $l$ and number of rounds needed ($r=3$ for $l=d-1$, $r=1$ for $l=d$). It should be an interesting result to be announced.

**Weaknesses:**

I think overall the paper is not hard to follow, but the presentation can be better
- The contents on page 4 and 5 would be a lot easier to understand if there is a picture.
- The algorithms deserve a box highlighting each step either in description or pseudocode, though I personally prefer just words.

Another issue is that it seems knowing the distance profiles is sufficient for the isometry test. Then why do we have to converge to WL test? The time and space consumption of WL-test is non-trivial. Since we have the two point sets at our hand, just consider a simple algorithm calculating the profile tuple $(A, M_1, M_2, ...)$ and use the proposed reconstruction algorithm, what is wrong with this? I believe the space might be the same, but the time would be much better.

minor: line 268, $d$-tuple?.

Another suggestion on Table 1.  I think using $l>d$ WL test is no longer of interest, maybe you can replace the dots with a slash. You can consider different colors for previous work, this work, and open. Last, there could be general $d$.

**Questions:**

See above.

**Limitations:**

I would say the result itself is interesting, only that as a theory paper the technical novelty is quite limited, especially if this can be done even without WL-test. The connection to GNN is in a phase of "Good to know" but does not seem to have application implications.

---

> ### Author Rebuttal · Authors · 2023-08-08
>
> Reviewer's comment:  "I think overall the paper is not hard to follow, but the presentation can be better
> The contents on page 4 and 5 would be a lot easier to understand if there is a picture.
> The algorithms deserve a box highlighting each step either in description or pseudocode, though I personally prefer just words."
>
> RESPONSE:  see “GENERAL RESPONSE” in the Author Rebuttal box at the beginning.
>
> Reviewer's comment: "Another issue is that it seems knowing the distance profiles is sufficient for the isometry test. Then why do we have to converge to WL test? The time and space consumption of WL-test is non-trivial. Since we have the two point sets at our hand, just consider a simple algorithm calculating the profile tuple and use the proposed reconstruction algorithm, what is wrong with this? I believe the space might be the same, but the time would be much better."
>
>
> REPONSE: The reviewer is right in that the point of the paper is not to provide an efficient algorithm to test isometry between two given point clouds, but to provide a characterization of the expressive power of geometric GNN’s. In particular, it informs the practical decision related to the choice of parameters in the design of a MPGNN’s when working with d-dimensional point clouds:  order d and three layers are sufficient (as far as expressivity is concerned).

---

### Official Review · Reviewer_4J2w · 2023-07-03

**Soundness:** 4 excellent
**Presentation:** 3 good
**Contribution:** 4 excellent
**Rating:** 8
**Confidence:** 5

**Summary:**

The authors rigorously prove that applying the d-1 WL to the distance matrices of point clouds in d dimensions is complete: that is, two point clouds are related by an isometry (and relabeling of the points) if and only if they are not separated by the d-1 WL test

**Strengths:**

The questions the authors address is a fundamental theoretical question in the study of geometric machine learning. They give a rigorous and non-trivial proof that essentially solves the question. Very good work.

**Weaknesses:**

No significant weaknesses.

**Questions:**

In lines 43-49 in the discussion of Hordan et al.: The authors of that paper discuss not only a 3-WL algorithm but also what the call in the abstract `a 2-WL-like algorithm', when d=3. To avoid confusion it may be helpful  to address this claim in the discussion somehow.

I had some issues with Section 6:
(a) Firstly, defining MPGNNs for point clouds has been discussed in various ways in previous works: in  [9]-[10] cited in the paper and in "On the expressive power of geometric graph neural networks" by Joshi et al which should be cited as well. It would make sense to discuss the relationships to the definitions there or at least mention they exist
(b) In the described MPGNN each x is given a `one hot encoding'. Since there are infinite possible x, how exactly is this accomplished?
(c) I was confused at why Corollary 6.1 was stated for (d-1)-MPGNN instead of d-MPGNN. Later I saw that this is explained in lines 75-80 and is due to the differences between WL and Folklore-WL but I think it should be retierated in Corollary 6.1
(d) Generally, I feel like this paper does a great job re the WL tests themselves, but the discussion of MPNNs is somewhat short and non-convincing, and perhaps it would be better just to ref to the other papers mentioned above which discussed these issues in more depth (at the authors discretion, I support acceptance either way)

I think the proof of Lemma 3.1 can be shortened and simplified. Once the equation in line 211 is established, you can immediately show that plugging it into the right hand side of (2), for both f(x) and f(y), gives the equality you want.

With the space the freed up, you could consider adding an illustration for the cone condition (lines 137-138)

Tiny comments:
Line 129: `Algorithm' should not be capitalized
Line 137: **line** on this line
Line 142: Initialization Data: instead of `it consists..' I would prefer something like `the initialization data consists..'





**Limitations:**

Yes

---

> ### Author Rebuttal · Authors · 2023-08-08
>
> Reviewer's comment: "In lines 43-49 in the discussion of Hordan et al.: The authors of that paper discuss not only a 3-WL algorithm but also what the call in the abstract `a 2-WL-like algorithm', when d=3. To avoid confusion it may be helpful to address this claim in the discussion somehow.”
>
> RESPONSE: Both algorithms in Hordan et al. explicitly use coordinates of the points. Thus, formally, they do not fall into our framework. However, a 3-WL algorithm of Hordan et al., after additional observations, can be used to show that 3-WL, as defined in our paper, is complete in R^3 after 2 iterations (we discuss this in Section 5 where we also improve their result by showing that 3-WL is complete in R^3 after 1 iteration). At the same time, we do not see how to convert the ‘2-WL-like’ algorithm of Hordan et al. into an algorithm in our setting. Nevertheless, we will mention their ‘2-WL-like’ algorithm with these remarks in the introduction.
>
>
> Reviewer's comment: “I had some issues with Section 6:
> (a) Firstly, defining MPGNNs for point clouds has been discussed in various ways in previous works: in [9]-[10] cited in the paper and in "On the expressive power of geometric graph neural networks" by Joshi et al which should be cited as well. It would make sense to discuss the relationships to the definitions there or at least mention they exist”
>
> RESPONSE: We were not aware of the paper by Joshi et al. It is certainly relevant and we will cite it. Other than that, we plan to completely rewrite this section as explained in point (d) below.
>
> Reviewer's comment: “(b) In the described MPGNN each x is given a `one hot encoding'. Since there are infinite possible x, how exactly is this accomplished?"
>
> RESPONSE: We were thinking of having a one-hot encoding only with respect to the “atomic types” of mutual distances that can be achieved in the given cloud of points S. There is a finite number of them. Hence, two tuples x and y in S achieve the same one-hot encoding if and only if they have the same atomic type of mutual distances.
>
>
> Reviewer's comment: “(c) I was confused at why Corollary 6.1 was stated for (d-1)-MPGNN instead of d-MPGNN. Later I saw that this is explained in lines 75-80 and is due to the differences between WL and Folklore-WL but I think it should be retierated in Corollary 6.1"
>
> RESPONSE: Sure, we can do that, but please see our response to the next point.
>
> Reviewer's comment: “(d) Generally, I feel like this paper does a great job re the WL tests themselves, but the discussion of MPNNs is somewhat short and non-convincing, and perhaps it would be better just to ref to the other papers mentioned above which discussed these issues in more depth (at the authors discretion, I support acceptance either way)"
>
> RESPONSE: After careful consideration, we believe that the reviewer is right. This section is too short and is not adding anything essentially new to the paper. In case we get accepted we will add a more succinct explanation saying that, by using previously established techniques, it is possible to construct efficient MPGNNs that achieve the expressive power of the WL test on clouds of points.
>
> Reviewer's comment: “I think the proof of Lemma 3.1 can be shortened and simplified. Once the equation in line 211 is established, you can immediately show that plugging it into the right hand side of (2), for both f(x) and f(y), gives the equality you want."
>
> RESPONSE: We thank the reviewer for the suggestion: indeed, it seems to make the proof shorter and simpler.
>
> Reviewer's comment: “With the space the freed up, you could consider adding an illustration for the cone condition (lines 137-138)"
>
> RESPONSE: That’s a good idea, thanks. We will definitely add such a figure in the final version of the paper.

---

> > ### Comment · Reviewer_4J2w · 2023-08-11
> >
> > I am happy with the reviewers answers. Thanks!

---

### Official Review · Reviewer_q73v · 2023-07-07

**Soundness:** 4 excellent
**Presentation:** 4 excellent
**Contribution:** 3 good
**Rating:** 7
**Confidence:** 3

**Summary:**

The expressive power of GNNs has long been a central topic in the GNN community. WL test serves as a fundamental algorithm in guiding designing expressive GNNs. For each k-WL, it has been shown that there always exist non-isomorphic graphs that cannot be distinguished by k-WL, and thus k-WL is incomplete for any k. However, things become different for geometric graphs, which are point clouds lying on a finite d-dimensional Euclidean space and the isomorphism is characterized by isometry. This paper makes a significant contribution by proving that k-WL is complete for distinguishing (k+1)-dimensional geometric graphs. Moreover, a constructive method shows that three-iteration suffices, which contrasts to a well-known result that there exist non-isomorphic graphs that cannot be distinguished by standard k-WL within o(n) iterations where n is the number of nodes. The authors further proved that k-WL can distinguish k-dimensional geometric graphs in only one iteration.

**Strengths:**

1. **Fundamental problem**. I believe characterizing the upper and lower bound of k such that k-WL is complete for distinguishing d-dimensional geometric graphs is a fundamental problem. This is due to a series of reasons.
   - First, k-WL is a very elegant algorithm and applies to geometric graphs straightforwardly.
   - Second, there have been debates whether equivariant architectures that use coordinate-wise information is necessary for learning geometric graphs, or purely distance information suffices. This paper answers this question timely and affirmatively.
Overall, the problem formulation is very *clean* and are likely to have decent impact in the geometric deep learning community.

2. **Strong theoretical result**. After going through the proof technique, I feel that the proof is non-trivial, despite presentation is very clear, well-organized, and rigorous. After checking several of the proofs, I am confident that the proofs are correct.
   - Related to prior works: to my knowledge, the theoretical result seems to be new. While I mainly focus on standard GNN expressivity theory, I have read several works related to this paper, such as Pozdnyakov & Ceriotti et al., Hordan et al., and Zhang et al..
   - Regarding completeness of theoretical result: this paper only gives upper bounds on the required k for distinguishing d dimensional geometric graphs. However, the bound is tight for 2 or 3 dimensional data. Other cases are highlighted in the limitation part and left as open problems. In this sense, the contribution seems to be sufficient for an acceptance.

3. **Great presentation**. This paper is very well-written and easy to read. The organization is great, the proof sketch is carefully written, and the proof in the Appendix is also well-written (I couldn't even find a typo). I really enjoy reading this paper.


**Weaknesses:**

1. This is mainly a theoretical paper, without any experimental evaluation. However, I understand that showing experimental result is not very necessary given this paper focuses on a fundamental theoretical problem.
2. Regarding theoretical results, I have several concerns and questions. While all of them are not major weaknesses, I would like the authors to answer these questions and revise the manuscript accordingly in the camera-ready version.
   - The authors focus on the setting where all points are distinct, i.e., S is a set rather than multiset. The proof in Line 146-157 requires such a condition. But I think similar result should hold when S is a multiset. The authors may add a brief remark to illustrate this point.
   - The authors wrote that Hordan et al. have proved that in the same setting, the geometric 3-WL test is in fact complete. However, I cannot find this result in their paper. Instead, they consider a different algorithm that requires the coordinates as input (although the output is invariant under isometric transformation). Moreover, they also considered higher dimensions. So could the authors give an explanation for the paragraph in Line 43-49 (if I miss something)?

3. The number of related work seems to be a bit insufficient. I can list several works which I think is relevant to this paper.
   - Martin Furer. Weisfeiler-Lehman Refinement Requires at Least a Linear Number of Iterations.

     This paper proved that, for any k, there exist non-isomorphic graphs that cannot be distinguished by standard k-WL within o(n) iterations where n is the number of nodes and can be distinguished by standard k-WL within $\Theta(n)$ iterations. Moreover, the same pair of graphs can be distinguished within much fewer iterations by increasing k. These results surprisingly parallels your results.
   - Bohang Zhang, et al.. Rethinking the Expressive Power of GNNs via Graph Biconnectivity.

     This paper proposed the expressive power of generalized distance WL test, which is basically the same as 1-WL in this paper while using different distance metrics.

4. Minor issue: in Line 54, the citation [12] seems to be wrong. Do you mean [13]?

**Questions:**

See the weakness part above.

**Limitations:**

The limitation has been clearly stated in the paper.

---

> ### Author Rebuttal · Authors · 2023-08-08
>
> Reviewer's comment: "This is mainly a theoretical paper, without any experimental evaluation. However, I understand that showing experimental result is not very necessary given this paper focuses on a fundamental theoretical problem."
>
> RESPONSE: Indeed, as the referee points out, the aim of the paper is to give theoretical guarantees and proofs. It is nevertheless important to point out, however, that relevant experiments have in fact been performed by other groups, such as described in (table 1, third column from the right, in) the preprint [Li, Z., Wang, X., Huang, Y., & Zhang, M. (2023), Is Distance Matrix Enough for Geometric Deep Learning? arXiv preprint arXiv:2302.05743, (version 4)]. We shall highlight these experiments further in the camera-ready version of the paper, in case it is accepted.
>
> Reviewer's comment: "The authors focus on the setting where all points are distinct, i.e., S is a set rather than multiset. The proof in Line 146-157 requires such a condition. But I think similar result should hold when S is a multiset. The authors may add a brief remark to illustrate this point."
>
> RESPONSE: We thank the referee for pointing this out. While some of the wordings of the proofs include reference to the hypothesis that S is a set, the proof strategy works without changes for the extension to the case S is a multiset. In case we get accepted, we shall add this extension of the theorem statements for the final version.
>
> Reviewer's comment: "The authors wrote that Hordan et al. have proved that in the same setting, the geometric 3-WL test is in fact complete. However, I cannot find this result in their paper. Instead, they consider a different algorithm that requires the coordinates as input (although the output is invariant under isometric transformation). Moreover, they also considered higher dimensions. So could the authors give an explanation for the paragraph in Line 43-49 (if I miss something)?"
>
> RESPONSE: Although the geometric 3-WL algorithm of Hordan et al. uses coordinates as inputs, it can be turned into a proof that 3-WL, as defined in our paper (that only uses pairwise distances) is complete in R^3 after 2 iterations. We discusse this in more detail at the beginning of Section 5, and we plan to add a reference to this in the introduction (lines 43-49) for a final version of the paper. We thank the reviewer for pointing out that Hordan et al. also consider higher dimensions. We will add a remark that the algorithm of Hordan et al., although it explicitly uses coordinates, can be turned into a proof (modulo the Barycenter lemma that we establish in our paper) that d-WL is complete in R^d after 2 iterations.
>
> Reviewer's comment: "The number of related work seems to be a bit insufficient. I can list several works which I think is relevant to this paper:
>
> - Martin Furer. Weisfeiler-Lehman Refinement Requires at Least a Linear Number of Iterations.
> This paper proved that, for any k, there exist non-isomorphic graphs that cannot be distinguished by standard k-WL within o(n) iterations where n is the number of nodes and can be distinguished by standard k-WL within $\Theta(n)$ iterations. Moreover, the same pair of graphs can be distinguished within much fewer iterations by increasing k. These results surprisingly parallels your results.
>
> - Bohang Zhang, et al.. Rethinking the Expressive Power of GNNs via Graph Biconnectivity.
> This paper proposed the expressive power of generalized distance WL test, which is basically the same as 1-WL in this paper while using different distance metrics."
>
> RESPONSE: We were not aware of the above papers. They are certainly very relevant and we will cite them, and we thank the referee for pointing them out to us.
>
> Reviewer's comment: "Minor issue: in Line 54, the citation [12] seems to be wrong. Do you mean [13]?
>
> RESPONSE: Indeed, the referee is correct, we will change this citation accordingly.

---

> > ### Comment · Reviewer_q73v · 2023-08-21
> > **Thank you**
> >
> > Thank you for your thoughtful response. My concerns have been addressed and I would be happy to see this paper accepted.

---

### Official Review · Reviewer_Z3UF · 2023-07-07

**Soundness:** 3 good
**Presentation:** 3 good
**Contribution:** 3 good
**Rating:** 6
**Confidence:** 1

**Summary:**

The paper addresses the question of testing the existence of a one-to-one application between two point clouds such that the distances are preserved. This information can be used, for example, to build better structured graph neural network architectures to optimize their performance. The starting point is that the distances between elements of a point cloud can be used to label the edges of a graph connecting them. Hence, the problem boils down to the detection of isometry between two graphs, for which the Weisfeiler-Lehman test is the classic tool. However, the latter makes it possible to conclude if two clouds are not isometric, but not necessarily that they are, depending on the computation cost. This paper sheds light on the issue in terms of the size of the ambient space

**Strengths:**

The document is fairly well written and the contributions are clearly stated. Since the results are theoretical in nature, they essentially consist of a succession of proofs. But they seem to be rigorously executed.
Not being a specialist in the field, I can't really appreciate the impact of such a theoretical result. This brings me to my next comment on the weaknesses.

**Weaknesses:**

The stated result, i.e. the ability to answer affirmatively if isometry between point clouds is proven from the WL test, is in itself very interesting. It squarely fits into "understanding ML results".
However, the authors stressed the importance of their results in improving the design of neural net architecture. This is unfortunately not clarified and it should be nice if the authors could comment more on that.

**Questions:**

The proofs are sometimes very verbose and would benefit from geometric illustration rather than prosaic text. The proof of lemma 3.1 (which seems connected to Konig-Huygens theorem) can easily go in the appendix to save space. As it is, I find it difficult to follow the entire logic of the proofs beyond a line-by-line follow-up.

The corollary 6.1 seems constructive. It woulds be helpful to describe the algorithm in a pseudo-code and maybe illustrate the proposed MPGNN.

I believe this will make the paper more accessible to a wider audience.
Since I don't have the necessary knowledge, I can hardly judge the impact and practical importance of such a result.

---

> ### Author Rebuttal · Authors · 2023-08-08
>
> Reviewer's comment: "the authors stressed the importance of their results in improving the design of neural net architecture. This is unfortunately not clarified and it should be nice if the authors could comment more on that."
>
> RESPONSE:   What we mean is simply that, when working with cloud points in dimension d, our results inform the decisions regarding the choice of parameters of the geometric GNN: as far as expressivity is concerned, order d and three layers are sufficient.
>
>
> Questions:
> The proofs are sometimes very verbose and would benefit from geometric illustration rather than prosaic text. The proof of lemma 3.1 (which seems connected to Konig-Huygens theorem) can easily go in the appendix to save space. As it is, I find it difficult to follow the entire logic of the proofs beyond a line-by-line follow-up.
> The corollary 6.1 seems constructive. It woulds be helpful to describe the algorithm in a pseudo-code and maybe illustrate the proposed MPGNN. I believe this will make the paper more accessible to a wider audience. Since I don't have the necessary knowledge, I can hardly judge the impact and practical importance of such a result.
>
> RESPONSE:  see “GENERAL RESPONSE” in the Author Rebuttal box at the beginning.

---

### Author Rebuttal · Authors · 2023-08-08

GENERAL RESPONSE:  Many thanks for your comments and suggestions. Regarding presentation, we acknowledge that adding a figure to illustrate the main idea behind the proof of our main result, as well as giving a high-level description of the underlying algorithm in a framed box, will greatly improve the presentation. In case our paper is accepted we will definitely implement all these. We now proceed to respond to the individual comments.

---

### Decision · Program_Chairs · 2023-09-21

**Decision:**

Accept (poster)

**Comment:**

All the reviewers agreed that the paper is well-written the problematic is important and the contributions are significant. They have also raised some concerns about improving the clarity and missing citations; however, they are rather minor concerns. Hence, I am recommending an acceptance. Please implement all the promised changes in the camera-ready version.